# Investigating Morphological Changes of T-lymphocytes after Exposure with Bacterial Determinants for Early Detection of Septic Conditions

**DOI:** 10.3390/microorganisms10020391

**Published:** 2022-02-08

**Authors:** Kari Lavinia vom Werth, Theresa Wörmann, Björn Kemper, Philipp Kümpers, Stefanie Kampmeier, Alexander Mellmann

**Affiliations:** 1Institute of Hygiene, University Hospital Münster, 48149 Münster, Germany; Karilavinia.Vomwerth@ukmuenster.de (K.L.v.W.); twoerman@uni-muenster.de (T.W.); Stefanie.Kampmeier@ukmuenster.de (S.K.); 2Biomedical Technology Center of the Medical Faculty, University of Münster, 48149 Münster, Germany; bkemper@uni-muenster.de; 3Department of Medicine D, Division of General Internal and Emergency Medicine, Nephrology and Rheumatology, University Hospital Münster, 48149 Münster, Germany; Philipp.Kuempers@ukmuenster.de

**Keywords:** sepsis, T-lymphocyte, cell morphology, digital holographic microscopy

## Abstract

Sepsis is a leading cause of morbidity and mortality, annually affecting millions of people worldwide. Immediate treatment initiation is crucial to improve the outcome but despite great progress, early identification of septic patients remains a challenge. Recently, white blood cell morphology was proposed as a new biomarker for sepsis diagnosis. In this proof-of-concept study, we aimed to investigate the effect of different bacteria and their determinants on T-lymphocytes by digital holographic microscopy (DHM). We hypothesize that species- and strain-specific morphological changes occur, which may offer a new approach for early sepsis diagnosis and identification of the causative agent. Jurkat cells as a model system were exposed to different *S. aureus* or *E. coli* strains either using sterile determinants or living bacteria. Time-lapse DHM was applied to analyze cellular morphological changes. There were not only living bacteria but also membrane vesicles and sterile culture supernatant-induced changes of cell area, circularity, and mean phase contrast. Interestingly, different cellular responses occurred depending on both the species and strain of the causative bacteria. Our findings suggest that investigation of T-lymphocyte morphology might provide a promising tool for the early identification of bacterial infections and possibly discrimination between different causative agents. Distinguishing gram-positive from gram-negative infection would already offer a great benefit for the proper administration of antibiotics.

## 1. Introduction

Currently, sepsis is defined as a dysregulated host response to infection leading to life-threatening organ dysfunction [1]. An estimated 49 million cases and 11 million sepsis-related deaths were reported for 2017 [2]. Due to the very heterogeneous nature of this syndrome, affected by both pathogen and host factors, diagnosis remains a major challenge. Identifying sepsis as early as possible is crucial for adequate and targeted treatment to improve patient outcomes, as delayed or inappropriate treatment is associated with decreasing survival rates [3,4,5].

Despite the huge global burden and extensive research, blood culture sampling, a technique with marked limitations, remains the gold standard method for diagnosis and species identification of bloodstream pathogens, which is essential to initiate a targeted antimicrobial therapy if necessary. Up to 30% of culture results are false negative [6] and it takes 24–72 h to get the results, leading to delayed treatment initiation.

As an alternative, a range of sepsis biomarkers like acute phase proteins and cytokines are currently under investigation. However, to date, no single biomarker is available that can reliably predict subsequent organ failure [7,8]. Procalcitonin (PCT) and C-reactive protein (CRP) are amongst the most widely studied protein biomarkers. Both increase during sepsis but also in other, non-infective conditions and therefore lack specificity [8,9]. During the last years, white blood cell morphology has been proposed as a novel biomarker [10]. In recent studies, classical hematological analyzers were used to determine cell volume, content (conductivity), and granularity (scatter; VCS) parameters [11,12]. Especially mean cell volumes of monocytes and neutrophils proved to be helpful parameters in distinguishing septic patients from healthy controls [11,13,14]. Additionally, morphologic parameters might be useful in the management of sepsis by tracking the efficacy of treatment [10,12].

To study cells or tissues in the context of various diseases, quantitative phase imaging [15] with digital holographic microscopy (DHM) [16] is a relatively new approach and provides a promising tool for clinical diagnostics [17,18,19,20,21,22,23]. This technique is a variant of quantitative phase microscopy [16,24,25] where the interference pattern of an object illumination wave and an undisturbed reference wave is recorded for numerical reconstruction of phase contrast images. It requires neither artificial staining nor labeling and only low-intensity illumination is used ensuring minimal invasiveness during measurements. Therefore, DHM can be applied for long-term observation of living cells and allows for observation of dynamic morphological and physical changes like area, shape, refractive index, or phase contrast. Thereby different cellular events can be quantified, such as proliferation [26], migration [27], differentiation [28,29], or death [30].

In this study, we want to investigate the effect of various bacterial determinants on T-lymphocytes using time-lapse DHM. T-lymphocytes, as part of the adaptive immune response, display a specific action towards particular invading agents, thus we expected distinct morphological changes depending on the pathogen. Previous studies revealed different cytotoxic reactions of endothelial cells after exposure to *Escherichia coli* Shiga toxin 1 (Stx1) or Stx2, indicating specific cellular responses depending on the virulence factor [31]. Furthermore, gram-negative as well as gram-positive bacteria release membrane vesicles (MVs) that are involved in various biological processes, including both physiological (e.g., communication, gene transfer) and pathological (e.g., immunomodulation, toxin transport) functions [32]. These MVs can also play an important role in pathogen–host interaction. They contain a broad range of pathogen-associated molecular patterns (PAMPs), like proteins, toxins, DNA, or RNA, that have the ability to interfere with the immune system [33]. We therefore focused not only on living bacteria but also analyzed the effect of sterile culture supernatants and MVs derived from *Staphylococcus aureus* and *E. coli*, the two most common gram-positive and gram-negative bacterial pathogens, respectively, in septic patients [6,34]. We hypothesize that different bacterial species and even strains from the same species as well as different bacterial determinants lead to specific morphological changes in T-lymphocytes that can be detected by DHM. Ultimately, we aim to test this approach, i.e., examination of white blood cell morphology using DHM, in an in vivo model to investigate its possible application in the clinical setting. It might provide the opportunity to gain rapid information about the infecting agent of patients with a suspected bacterial infection.

## 2. Materials and Methods

### 2.1. Cell Culture

Jurkat cells (TIB-152; ATCC, Manassas, USA), a human T-lymphocyte cell line, were cultivated in Roswell Park Memorial Institute 1640 medium (RPMI; Lonza, Cologne, Germany) supplemented with 10% fetal calf serum (FCS; PAA, Pasching, Austria) and 2 mM Ultraglutamine (Lonza) in a humidified atmosphere at 37 °C and 5% CO_2_. Cell density was maintained at 8 × 10^4^ to 1 × 10^6^ viable cells/mL by passaging the cells every 3–4 days.

### 2.2. Bacterial Strains and Culture Conditions

The following species were used in this study: *S. aureus* (strains 6850 (clinical sepsis isolate), USA300 (prototypic community-acquired *S. aureus*), LS1 (laboratory strain from infective arthritis in mice), Newman (clinical isolate from osteomyelitis), SH1000 (laboratory strain), Cowan I (human septic arthritis), and ST398 (prototypic livestock-acquired *S. aureus*)) and *E. coli* (strains 536 (prototypic uropathogenic *E. coli* [UPEC]), IHE3034 (newborn meningitis-associated *E. coli* [MNEC]), and MG1655 (commensal *E. coli*)) (Table 1).

Prior to the experiments, the bacteria were plated on lysogeny broth (LB) agar (Roth, Karlsruhe, Germany) plates and incubated overnight at 37 °C. Single colonies were taken to inoculate overnight cultures in LB medium (Roth) and bacteria were grown at 37 °C with shaking at 180 rpm.

### 2.3. Preparation of Bacterial Membrane Vesicles and Culture Supernatants

Bacterial MVs from gram-negative as well as gram-positive strains were prepared as previously described [45]. Briefly, bacterial strains were grown overnight in LB medium at 37 °C with shaking at 180 rpm. After removal of bacteria by centrifugation (5600× *g*, 20 min, 4 °C) the supernatants were sterile filtered, and MVs were sedimented by ultracentrifugation (235,000× *g*, 2 h, 4 °C) in a 45 Ti rotor (Beckman Coulter, Krefeld, Germany). Subsequently, MVs were resuspended in 20 mM TRIS-HCl (pH 8.0) and stored at 4 °C. Protein concentrations were determined using the Pierce BCA Protein Assay Kit (ThermoFisher Scientific, Dreieich, Germany) following the manufacturer’s instructions. For treatment of cells, MV suspensions were diluted in cell culture medium to a final concentration of 20 µg/mL. MV preparations were used for experiments within 3 weeks after preparation.

To prepare sterile culture supernatants, bacteria were grown in RPMI medium overnight at 37 °C with shaking at 180 rpm. Cultures were centrifuged (5600× *g*, 5 min, 4 °C) and supernatants were passed through 0.22 μm pore-size filters (Corning, Wiesbaden, Germany) to remove the remaining bacteria. For every experiment, sterile culture supernatants were freshly prepared and added to the cell culture medium in the indicated dilutions ranging from 5% to 25% (*v*/*v*).

### 2.4. Bacterial Infection

For infection of Jurkat cells with living bacteria, all strains were grown overnight in LB medium. Bacterial cultures were centrifuged (5600× *g*, 5 min, RT), washed with PBS (Lonza), and adjusted to an optical density at 600 nm (OD600) of 1 in PBS (corresponding to 5 × 10^8^ colony forming units (CFU)/mL for *S. aureus* and 8 × 10^8^ CFU/mL for *E. coli*, respectively). Jurkat cells at a density of 1.3 × 10^5^ cells/mL were infected with a multiplicity of infection (MOI) of 5 or 10. For short-term DHM measurements, time-lapse microscopy was started directly after infection without further treatment. For long-term experiments, cells were pelleted after 4 h (50× *g*, 10 min, RT) and bacteria were removed from the supernatant by centrifugation at 5600× *g* for 5 min. The supernatant was then added back to the cells and supplemented with 2 µg/mL Lysostaphin (Sigma-Aldrich, Darmstadt, Germany) or 50 µg/mL Gentamycin (Sigma-Aldrich) for *S. aureus* or *E. coli* infected cells, respectively, to prevent overgrowth of bacteria. Time-lapse microscopy was started 5 h post-infection (p.i.). Mock-infected cells were treated equally but without the addition of bacteria and served as control.

### 2.5. Digital Holographic Microscopy (DHM)

Time-lapse DHM of living cells was performed using an inverted microscope (IM-3, Optika, Ponteranica, Italy) with a motorized stage (Märzhäuser, Wetzlar, Germany) and an attached fiber-optic DHM module that is placed inside an incubator (MCO-20AIC, Panasonic, Osaka, Japan) to ensure stable temperature and CO_2_ conditions. This setup is based on a concept described by Kemper et al. [46]. The coherent light source used to acquire digital off-axis holograms was a single longitudinal mode laser (Cobolt 06-DPL, λ = 532 nm, 25 mW; Cobolt AB, Solna, Sweden). Exposure time was set to 0.21 ms to minimize phototoxicity and to ensure interferometric stability during hologram recording. Holograms were captured with a complementary metal-oxide-semiconductor (CMOS) sensor (UI-3260CP-M-GL Rev.2; IDS Imaging Development Systems, Obersulm, Germany). To reduce coherence and pathogen-induced image disturbances, the sample illumination light was modulated by an electrically focused tunable lens (ETL) [47] and a series of digital off-axis holograms were recorded. The holograms were acquired automatically every 3 min over a period of up to 24 h.

Quantitative phase images were numerically reconstructed as previously described [48] using custom-built Python 3 based software developed at the Biomedical Technology Center Münster, Germany. After numerical reconstruction, averaged quantitative phase images were calculated from every hologram series that was acquired with modulated object illumination by the ETL [47]. Segmentation of the resulting images and particle analysis of morphological parameters and phase contrast changes was performed using FIJI software [49] (version 2.3.0/1.53f51). A schematic representation of the DHM setup is provided in Appendix A, and for detailed information on the data analysis procedure refer to Appendix A.

For DHM measurements, Jurkat cells were adjusted to a density of 1.3 × 10^5^ cells/mL and treated with bacterial MVs, sterile culture supernatants, or living bacteria. Untreated or mock-infected cells served as control. The cell suspension was transferred to 4-well Ph+ µ-slides (ibidi, Gräfelfing, Germany) and sealed with anti-evaporation oil (ibidi). Three different positions were recorded for every treatment and three independent time-lapse measurements were performed for each experiment. For data analysis, cells recorded at the three different positions of the same treatment were pooled. Different morphological parameters for single cells (e.g., area, perimeter, circularity, etc.) and phase contrast values (e.g., mean, modal, min, max) were determined as an average of about 60 cells per time point. Results are presented as mean ± SD of three or more biological replicates.

## 3. Results

### 3.1. Strain-Dependent Morphological Changes of Jurkat Cells Exposed to S. aureus Supernatant or MVs

Bacteria secrete a large number of proteins and molecules that have the potential to interfere with the host response [50]. It was shown that these soluble factors have different effects on host cells in contrast to living bacteria [51]. Therefore, we first examined the response of T-lymphocytes to sterile culture supernatant of different *S. aureus* strains by time-lapse DHM. As cellular responses upon infection also depend on the infecting *S. aureus* strain [52], we included strains from different origins that are known to have varying degrees of cytotoxicity. Indeed, we were able to identify distinct patterns of cellular morphological changes that can be divided into three different phenotypes (Figure 1). Supernatants of *S. aureus* strains 6850, USA300, and LS1 induced rapid and strong morphological alterations in Jurkat cells that could be quantified using DHM as immediate cellular mean phase contrast decrease and transient reduction of circularity. In contrast, supernatants of strains SH1000 and ST398 led to an early increase of cellular mean phase contrast within the first five hours, which then gradually decreased over time. Temporary reduction of the circularity occurred similar to strains USA300 and LS1 but shrinkage of the cells, i.e., the area was considerably stronger (Figure 1). Jurkat cells exposed to the supernatant of strains Newman or Cowan I did not exhibit any changes in phase contrast or cell shape as compared to untreated control cells.

Since bacteria release MVs that might play a role in disease progression [53,54], we aimed to investigate the effect of MVs only on T-lymphocytes. Therefore, Jurkat cells were treated with MVs isolated from the supernatant of *S. aureus* cultures. Time-lapse DHM measurements revealed a rapid cellular response after treatment with *S. aureus* MVs (Figure 1a and Appendix A). All strains, with the exception of Cowan I, led to a decrease of cellular mean phase contrast, circularity, and cell area. However, changes in the cell shape, i.e., circularity, predominantly occurred within the first three hours and recovered over time. *S. aureus* strains 6850, USA300, LS1, SH1000, and ST398 showed comparable morphological changes whereas the cellular response of strain Newman had the same pattern but occurred later in time. In line with the previous results obtained with culture supernatant, strain Cowan I did not have any effect on phase contrast or cell shape.

To investigate whether there are also concentration-dependent differences, Jurkat cells were exposed to different amounts of *S. aureus* 6850 supernatants. Lower concentrations of 5% or 10% (*v*/*v*) induced comparable cellular morphological changes with an early decrease of mean phase contrast and slight reduction of cell area (Figure 2). In contrast, treatment with 25% (*v*/*v*) supernatant led to a rapid disintegration of cell integrity as indicated by a marked reduction of cell area, circularity, and mean phase contrast.

### 3.2. Differential Responses of Jurkat Cells Infected with Living S. aureus

To investigate not only the effect of soluble factors secreted by bacteria but also the living pathogen, Jurkat cells were infected with *S. aureus* (MOI 5 or 10). In preliminary tests, we could not observe any changes during the first hours after infection, therefore only long-term measurements starting at 5 h p.i. are shown here. Surprisingly, only infection with strain 6850 caused changes in cell morphology while cells infected with strain USA300 or Newman did not differ from mock-infected control cells (Figure 3a and Appendix A). However, for strain 6850 we could observe different morphological changes depending on the bacterial load (Figure 3b,c). With an increasing MOI, a greater reduction of circularity and mean phase contrast occurred while cell shrinkage was similar between both treatments.

### 3.3. Different Effects of E. coli Culture Supernatant and MVs on Jurkat Cells

Besides *S. aureus*, one of the predominant causes of sepsis is gram-negative *E. coli* [34]. Therefore, we aimed to include also *E. coli* strains to investigate whether T-lymphocytes exhibit morphological changes different from *S. aureus* treatment. Again, we first exposed Jurkat cells to sterile culture supernatant, or bacterial MVs derived from different *E. coli* strains and major differences could be observed. Treatment with supernatant of strain 536 immediately induced a decrease in cellular phase contrast, circularity, and cell area (Figure 4a,c). In contrast, Jurkat cells exposed to the supernatant of IHE3034 showed a completely different response. Here, only a delayed cellular response (after 15 h) could be determined with a slight increase of the mean cellular phase contrast while circularity, as well as cell area, declined. Using supernatant of the commensal strain MG1655 did not induce any changes in comparison to the untreated control cells.

Interestingly, unlike *S. aureus*-derived MVs, we could not observe any effect when exposing Jurkat cells to *E. coli* MVs (Figure 4b and Appendix A).

### 3.4. Strain-Dependent Morphological Changes of Jurkat Cells Infected with Living E. coli

Two different experimental setups were used to investigate the effect of living *E. coli* on Jurkat cells. On the one hand, bacteria were removed after 4 h by centrifugation and antibiotic treatment to prevent bacterial overgrowth during long-term observation. In this case, DHM was started 5 h p.i. On the other hand, DHM measurement was started directly after infection to capture early morphological changes. The cellular response upon infection with *E. coli* strongly depended on the strain. Starting DHM directly after infection, strain 536 led to early morphological changes beginning around 1 h p.i. with a reduction in mean phase contrast and circularity (Figure 5). Cell area first slightly increased and afterward started to decline.

In contrast, *E. coli* IHE3034 and MG1655 did not induce such rapid effects. For these strains, a cellular effect was only observed in long-term measurements (Figure 6). Eight h after the addition of the bacteria cell morphology starts to change with a constant decrease of cell area and circularity over time. Here, however, we could not determine any differences in mean phase contrast when compared to mock-infected control cells. Strain 536 was excluded from the long-term measurements since Jurkat cells infected with these bacteria underwent cell death prior to the starting point of 5 h p.i. (Figure 5).

## 4. Discussion

Cell morphology of white blood cells changes under septic conditions, thereby providing a promising approach to distinguish sepsis from local infection or healthy controls [12,13,55]. However, most of the studies in this field were performed using classical hematology analyzers and focused on monocytes and neutrophils while morphological changes of T-lymphocytes were hardly described. Lymphocytes are part of the adaptive immune response; thus we speculated that these cells exhibit a more diverse response toward different bacterial species and strains compared to the non-specific, innate immune system. Therefore, we aimed to investigate the cellular response of T-lymphocytes under various experimental septic conditions by time-lapse DHM. We included different *S. aureus* and *E. coli* strains as these species are frequently isolated from septic patients [6,34]. The number of circulating bacteria in the blood is often low [56,57,58], we therefore treated cells not only with living pathogens but also sterile culture supernatants to analyze the effects of secreted virulence factors. Since host response to infection amongst other factors depends on the infecting agent [52], we included strains, which have various degrees of cytotoxicity. Indeed, we observed strain-dependent morphological changes of Jurkat cells after treatment with sterile culture supernatant. In this study, changes in single-cell area, cellular circularity, and mean phase contrast per cell are presented, since these parameters exhibited the most pronounced effects. *S. aureus* strains ST398 and SH1000 induced an early increase in mean phase contrast, reaching the maximum around 60 to 120 min with a subsequent decline (Figure 1). The mean phase shift is proportional to the cellular dry mass, i.e., the non-aqueous content, and the cell area [59,60]. A decreasing phase shift might be an indicator of cell death processes [61]. If the cell membrane is ruptured, intracellular content is released leading to a decrease of dry mass and therefore phase contrast. Furthermore, cell area strongly reduced over time, also indicating cell death [61]. The highly cytotoxic strains 6850, USA300, and LS1 in contrast led to an immediate and strong decrease of mean phase contrast while cell size only slightly declined. However, we cannot exclude a preceding increase in mean phase contrast, as seen for the aforementioned strains. Due to the DHM setup, we were not able to record the first 10 to 15 min after treatment of cells. Nevertheless, cell morphology also differed at later time points indicating distinct cellular responses depending on the bacterial strain.

In addition to free proteins and enzymes, bacteria release MVs that protect their cargo against degradation related to environmental conditions (e.g., pH) or enzymatic actions [62]. Pathogenic bacteria use MVs to transport a broad range of PAMPs and to interact with the host, thereby contributing to disease progression [53,54,63]. According to that, MVs isolated from the supernatant of *S. aureus* cultures induced a strong response in T-lymphocytes (Figure 1a). As expected and in line with previous results [52], we could not observe any effects of the non-cytotoxic strain Cowan I. All other strains, besides Newman, induced comparable morphological changes, namely a rapid reduction of mean phase contrast and constantly decreasing cell area. This behavior might be due to our experimental conditions, where we adjusted the protein concentration of MV suspensions. A recent study described that MVs derived from different *S. aureus* strains share a highly conserved core proteome but the numbers of MVs produced might be strained specific [64]. Therefore, since we used the same protein concentration of MVs across all strains, we observed similar effects of MVs derived from different *S. aureus* strains. For treatment with supernatant, in contrast, we used the same volume for each strain, but did not quantify protein concentrations. Presumably, the supernatants contain different amounts of virulence factors and hence cellular responses were more heterogeneous compared to MV treatment. These findings suggest that the diverse morphological changes rather come from differences in toxin quantity than the type of virulence factor. In line with that, we observed a concentration-dependent increase of the cytotoxic effect when Jurkat cells were treated with *S. aureus* 6850 supernatant (Figure 2).

Surprisingly, we did not observe such strong cellular responses when Jurkat cells were infected with living bacteria. Only infection with strain 6850 induced morphological changes that were comparable to the effects after treatment with supernatant or MVs (Figure 3a). Again, the extent of the reaction was concentration-dependent, as a higher MOI led to more pronounced cellular changes (Figure 3b,c). It can also be assumed that strains less cytotoxic than 6850 could induce a cellular response when the MOI is further increased. Different degrees of invasiveness might also contribute to the distinct effects. The invasion of *S. aureus* into host cells depends on both the strain and cell type [52]. Since we removed extracellular bacteria after 4 h by antibiotic treatment in our experimental setup, the following processes are largely influenced by intracellular bacteria. Interestingly, treatment with sterile determinants, i.e., MVs and supernatant, led to a much stronger response than living bacteria. We assume that this is a concentration-dependent effect. Treatment with MVs or supernatant directly exposes the cells to a broad range of virulence factors, such as toxins, that can rapidly induce cell death. In contrast, for living bacteria, it takes some time for replication and secretion of virulence factors until the concentration is high enough to induce a cellular response. Most likely, the use of higher bacterial loads for infection would induce stronger or earlier morphological changes. Furthermore, due to the experimental setup, bacteria needed to be removed for DHM measurements, which might also counteract the cellular response.

For early sepsis diagnosis and initiation of antibiotic treatment, distinguishing gram-positive from gram-negative infection would be of great benefit as treatment with pathogen-specific antimicrobials could be started immediately. *E. coli* is one of the most common organisms isolated from gram-negative sepsis [34]. We therefore repeated the DHM measurements with different *E. coli* strains to get a first impression if not only strain-specific but also species-specific differences occur. Unlike MVs secreted by *S. aureus*, *E. coli* derived MVs did not exhibit any effect on Jurkat cells during the first 24 h (Figure 4b). At least the pathogenic strains 536 and IHE3034 induced morphological changes during longer incubation periods up to 72 h (data not shown). These findings indicate major differences in cytotoxicity of MVs secreted by different bacterial species. In line with the reduced virulence potential of secreted determinants of *E. coli* in comparison to *S. aureus*, sterile culture supernatant derived from *E. coli* was less harmful to Jurkat cells compared to *S. aureus*. We had to apply a much higher concentration of 25% (*v*/*v*) supernatant to induce a cellular response and still, morphological changes varied greatly depending on the strain (Figure 4a). Treatment with supernatant of UPEC strain 536 induced a rapid decrease of mean phase contrast, circularity, and area, similar to *S. aureus* strain 6850 at the same concentration (Figure 2). In contrast, cellular changes after exposure to the supernatant of MNEC strain IHE3034 occurred later and showed a completely different pattern. Here cell area and circularity declined while mean phase contrast increased (Figure 4a). Similar differences could be observed after infection with living bacteria. *E. coli* strain 536 led to a rapid induction of morphological changes within the first 2 h after infection, reflected by a decrease of mean phase contrast and circularity (Figure 5). Strains IHE3034 and MG1655 did not affect cell integrity during this early stage of infection but induced a later response (Figure 6). Mean phase contrast remained constant while cell area and circularity continually declined. One explanation of these major differences among the *E. coli* strains investigated could be related to their different infection routes and ability to adapt to different environments [65]. Interestingly, direct contact of cells with living *E. coli* resulted in the strongest morphological changes, even stronger than with *S. aureus*. Based on our data we can only speculate about the underlying cause. Since the commensal strain MG1655 also induced a comparable response, it might come from a global factor (like LPS) rather than specific virulence factors.

Overall, we were able to determine morphological changes of T-lymphocytes after exposure to different bacteria or bacterial determinants using DHM as a label-free imaging technique. Interestingly, we could demonstrate inter- and intra-species differences. These data provide a basis for future in vivo experiments, i.e., the analysis of patient samples. Additionally, future studies may investigate the ability of this DHM technique to identify patients that are at high risk to develop a subsequent organ failure and to predict their outcome.

However, this study has several limitations. So far, we tested a limited number of bacterial species and strains using a cell culture model. It is very likely that our in vitro conditions can only partially reflect the highly heterogeneous septic responses in patients. Our findings thus have to be confirmed by in vivo analyses. Previous studies using hematology analyzers described significant differences in VCS parameters in monocytes and neutrophils of septic patients and healthy controls [10,11,12,13,14]. We therefore assume that we will be able to see comparable effects in T-lymphocytes. Nevertheless, in these studies, it was only differentiated between septic and non-septic patients irrespective of the causative bacteria. Since our in vitro experiments revealed major differences depending on the strain and species, it will be interesting to see if we can transfer these findings to an in vivo system. Furthermore, in the present study, we only investigated morphological changes under experimental infectious conditions. We cannot exclude that a similar cellular response might also occur in sterile inflammation processes. This hypothesis could be further investigated by stimulation of T-lymphocytes with cytokines such as TNF or interleukins. Another limitation is the choice of treatment concentrations. We tried to use concentrations as low as possible to approximate the in vivo situation. It cannot be excluded that higher or even lower amounts would elicit different effects. Moreover, in the present study, we focused on T-lymphocytes as part of the adaptive immune system. Further studies might include innate immune cells, like monocytes and neutrophils. These cells presumably show different responses that might be more homogeneous compared to T-lymphocytes. Further limitations might result from the image analysis since we used a multi-step procedure and different settings may lead to changes in the absolute values. However, we assume that the relation of the different treatments to each other would remain the same. For better transparency and reproducibility, we provided detailed information on the image analysis in Appendix A.

Despite these limitations, our study offers new insights into the applicability of white blood cell morphology as a marker for early sepsis diagnosis and potentially even prediction of the outcome. We showed that label-free DHM can be used to distinguish species- and strain-specific effects on T-lymphocytes under in vitro conditions. Differentiation of gram-positive and gram-negative infection is of great importance for early administration of pathogen-specific antibiotics since inappropriate therapy results in decreased survival rates [5]. Our results indicate that T-lymphocyte morphology might be a promising tool for such point-of-care discrimination.

## Figures and Tables

**Figure 1 microorganisms-10-00391-f001:**
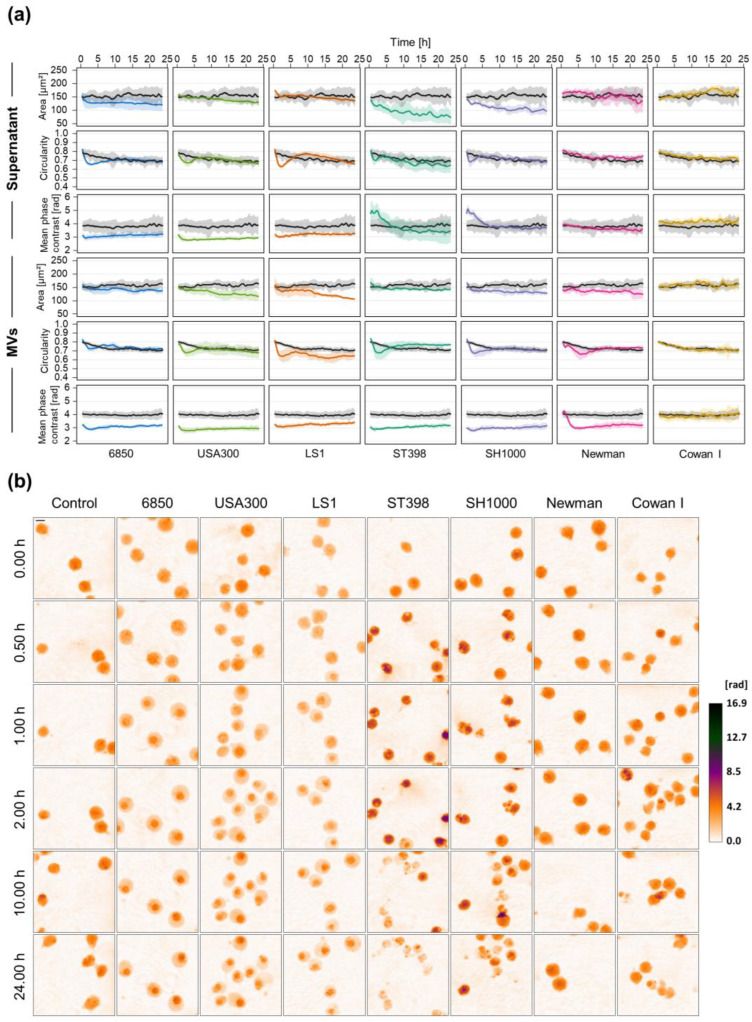
Cellular changes of Jurkat cells exposed to supernatant or MVs derived from different *S. aureus* strains. Jurkat cells were treated with 5% (*v*/*v*) sterile culture supernatant or 20 µg/mL MVs of different *S. aureus* strains. Untreated cells served as control. Cellular changes were monitored by time-lapse DHM. (**a**) DHM quantitative phase contrast images were analyzed for the average single-cell area (µm^2^), average circularity (AU), and mean phase contrast (rad) per cell. Results are presented as mean (lines) ± SD (shading) of at least three independent experiments. Controls are depicted in black while different colors designate the individual strains. For better visualization, curves were smoothed using moving averages with a window size of 15 measuring points. (**b**) Representative color-coded phase contrast images of Jurkat cells at indicated time points after the addition of bacterial supernatant. The scale bar corresponds to 10 μm. The calibration bar indicates phase contrast values in radian.

**Figure 2 microorganisms-10-00391-f002:**
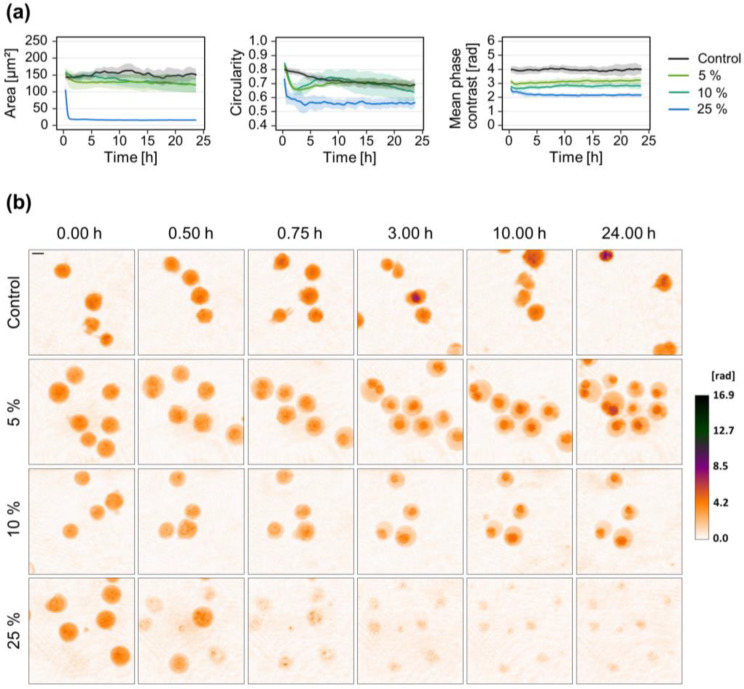
Concentration-dependent effect of *S. aureus* strain 6850 culture supernatant on Jurkat cell morphology. Jurkat cells were treated with different concentrations of sterile culture supernatant derived from *S. aureus* strain 6850. Untreated cells served as control. Time-lapse DHM was applied to investigate dynamic morphological changes. (**a**) DHM quantitative phase contrast images were analyzed for the average single-cell area (µm^2^), average circularity (AU), and mean phase contrast (rad) per cell. Results represent mean (lines) ± SD (shading) of at least three independent experiments. Controls are shown in black while different colors indicate different concentrations of supernatant. For better visualization, curves were smoothed using moving averages with a window size of 15 measuring points. (**b**) Representative color-coded phase contrast images of Jurkat cells at indicated time points after the addition of bacterial supernatant. The scale bar corresponds to 10 μm. The calibration bar indicates phase contrast values in radian.

**Figure 3 microorganisms-10-00391-f003:**
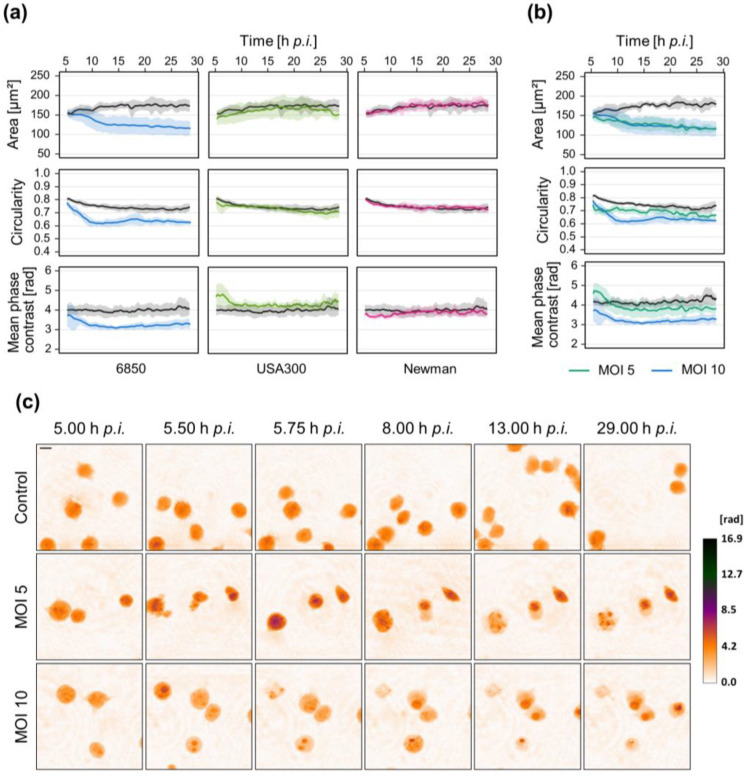
Morphological changes of Jurkat cells after infection with living *S. aureus* depend on strain and bacterial load. Jurkat cells were infected with different *S. aureus* strains (MOI 10; (**a**)) or different bacterial loads of *S. aureus* strain 6850 (**b**). After 4 h, bacteria were removed by centrifugation and Lysostaphin treatment (2 µg/mL) to prevent overgrowth. Time-lapse DHM was started at 5 h p.i. and mock-infected cells served as control. (**a**,**b**) DHM quantitative phase contrast images were analyzed for the average single-cell area (µm^2^), average circularity (AU), and mean phase contrast (rad) per cell. Results represent mean (lines) ± SD (shading) of at least three independent experiments. Controls are depicted in black while different colors mark the individual strains (**a**) or different bacterial loads (**b**). For better visualization, curves were smoothed using moving averages with a window size of 15 measuring points. (**c**) Representative color-coded phase contrast images of Jurkat cells infected with *S. aureus* strain 6850 at indicated time points after infection. The scale bar corresponds to 10 μm. The calibration bar indicates phase contrast values in radian.

**Figure 4 microorganisms-10-00391-f004:**
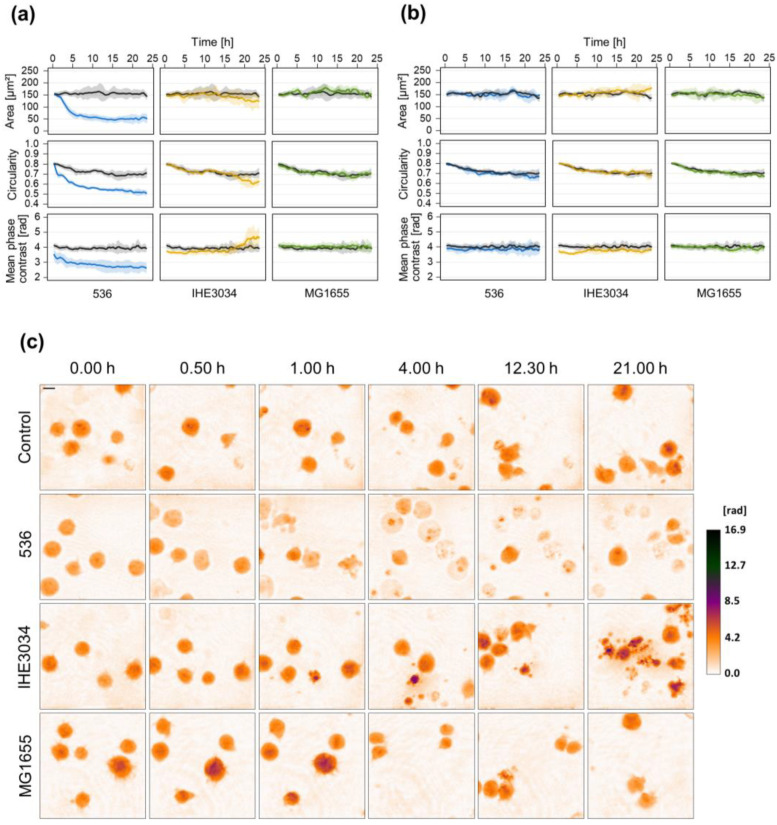
Culture supernatant but not MVs derived from different *E. coli* strains induce morphological changes in Jurkat cells. Jurkat cells were exposed to 25% (*v*/*v*) sterile culture supernatant (**a**) or 20 µg/mL MVs (**b**) of different *E. coli* strains. Untreated cells served as control. Time-lapse observation of morphological changes was carried out using DHM. (**a**,**b**) DHM quantitative phase contrast images were analyzed for the average single-cell area (µm^2^), average circularity (AU), and mean phase contrast (rad) per cell. Results are shown as mean (lines) ± SD (shading) of at least three independent experiments. Controls are presented in black while different colors designate the individual strains. For better visualization, curves were smoothed using moving averages with a window size of 15 measuring points. (**c**) Representative color-coded phase contrast images of Jurkat cells at indicated time points after the addition of *E. coli* culture supernatant. The scale bar corresponds to 10 μm. The calibration bar indicates phase contrast values in radian.

**Figure 5 microorganisms-10-00391-f005:**
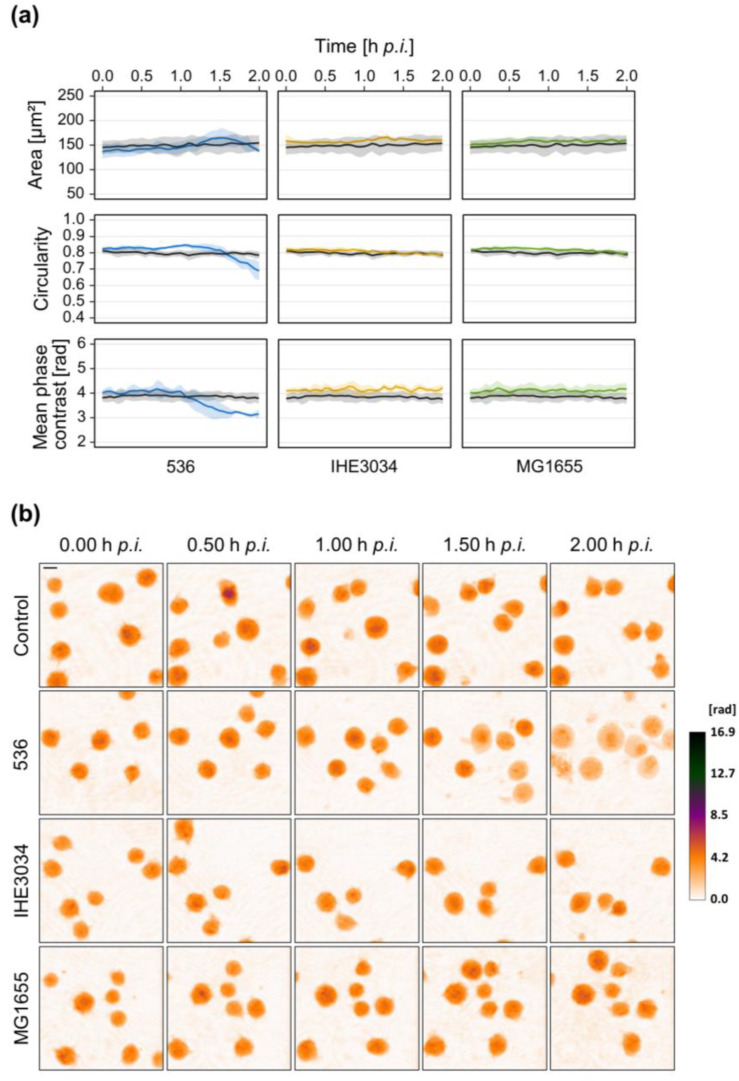
Early cellular changes of Jurkat cells infected with living *E. coli*. Jurkat cells were infected with different *E. coli* strains (MOI 10) and DHM was started directly after the addition of the bacteria. Mock-infected cells served as control. (**a**) DHM quantitative phase contrast images were analyzed for the average single-cell area (µm^2^), average circularity (AU), and mean phase contrast (rad) per cell. Results are shown as mean (lines) ± SD (shading) of at least three independent experiments. Controls are depicted in black while different colors represent the individual strains. For better visualization, curves were smoothed using moving averages with a window size of 2 measuring points. (**b**) Representative color-coded phase contrast images of Jurkat cells at indicated time points after the addition of bacteria. The scale bar corresponds to 10 μm. The calibration bar indicates phase contrast values in radian.

**Figure 6 microorganisms-10-00391-f006:**
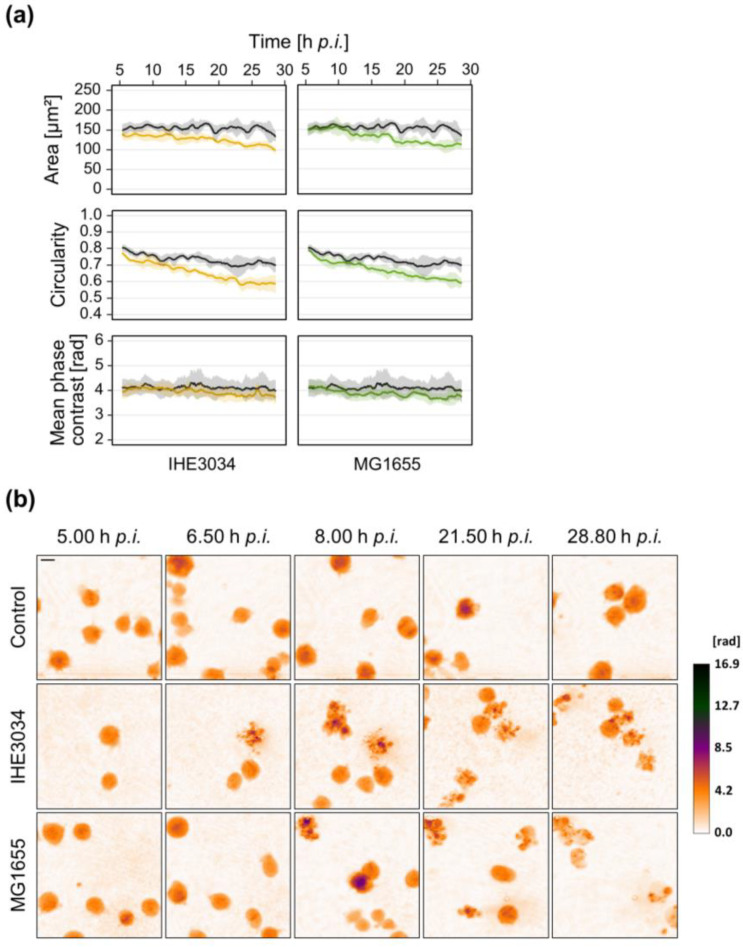
Morphological changes of Jurkat cells infected with *E. coli* strains IHE3034 or MG1655. Jurkat cells were infected with *E. coli* strains IHE3034 or MG1655 (MOI 10). After 4 h, bacteria were removed by centrifugation and Gentamicin treatment (50 µg/mL) to prevent overgrowth. Mock-infected cells served as control. DHM measurement was started at 5 h p.i. to monitor cellular morphological changes. (**a**) DHM quantitative phase contrast images were analyzed for the average single-cell area (µm^2^), average circularity (AU), and mean phase contrast (rad) per cell. Results represent mean (lines) ± SD (shading) of at least three independent experiments. Controls are shown in black while different colors designate the individual strains. For better visualization, curves were smoothed using moving averages with a window size of 15 measuring points. (**b**) Representative color-coded phase contrast images of Jurkat cells at indicated time points after infection. The scale bar corresponds to 10 μm. The calibration bar indicates phase contrast values in radian.

**Table 1 microorganisms-10-00391-t001:** Bacterial strains used in the present study.

Strain	Source	Reference
*S. aureus* 6850	Isolated from a patient with a skin abscess that progressed to *S. aureus* bacteremia, osteomyelitis, septic arthritis, and systemic abscesses	[35]
*S. aureus* USA300	Emerged as community-acquired MRSA in the USA and rapidly spread across the country to become the leading cause of MRSA infection also in healthcare settings	[36,37]
*S. aureus* LS1	Isolated from bacterial arthritis in mice	[38]
*S. aureus* ST398	Zoonotic MRSA frequently colonizing livestock animals but also causes human infections	[39]
*S. aureus* SH1000	Derivative of *S. aureus* NCTC 8325	[40]
*S. aureus* Newman	Isolated from human osteomyelitis	[41]
*S. aureus* Cowan I	Isolated from a patient with septic arthritis	ATCC 12598
*E. coli* 536	Isolated from a patient with urinary tract infection	[42]
*E. coli* IHE3034	Isolated from newborn meningitis	[43]
*E. coli* MG1655	*E. coli* K-12 wildtype	[44]

## Data Availability

Not applicable.

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
