# Peer review of "Investigating Morphological Changes of T-lymphocytes after Exposure with Bacterial Determinants for Early Detection of Septic Conditions"

_microorganisms, 2022, doi:10.3390/microorganisms10020391_

Round 1
Reviewer 1 Report
Authors investigated the morphological changes of T-lymphocytes stimulated with live bacteria or bacterial contents in vitro model.
It is very interesting, and important topic in sepsis research.
The followings should be commented and discussed.
1. The morphological change of Jurkat cells related to this study could be interpreted in caution. Sterile inflammation such as trauma, pancreatitis etc. could induce this morphological change. You could study this with stimulation Jurkat cells with some cytokines such as TNF. It should be at least commented in limitation section.
2. You observed morphological changes of Jurkat cell in longer period in S. aureus, but in short period in case of E.coli (Fig 3 vs. Fig 5, 6). Why did you make the observation time different between species?
3. In case of E.coli 536, you did not observe longer term period. What is the results of long term morphological change of Jurkat cells in E.coli 536. As you commented, if you would like to differentiate gram positive or negative infection with morphological change of T-cells, you should directly compare the same time points after infection, I think.
4. The title of figure 5 should be changed.
It was done with living E. coli, not only with living E.coli 536. So, 536 should be removed.
Reviewer 2 Report
In this study, the authors investigate changes in T-cell morphological characteristics in response to exposure for different bacterial species and strains using a Jurkat cell model. The perspective is the identification of new markers for the diagnosis of bacterial sepsis including differentiation between the underlying pathogen, with a view of facilitating more targeted early antibiotic treatment.
The topic is of interest and is clinically relevant. The paper is well written and the methods sufficiently described, barring a few comments minor comments. The authors' conclusions are in accordance with the presented data.
I have the following specific comments for the authors:
1) Introduction, page 2, lines 82-85: "Ultimately, this approach, i.e. examination of white blood cell morphology using DHM, should be the basis to provide rapid information about the infecting agent of patients with a suspected bacterial infection." This statement is perhaps a little optimistic given that your study is a proof-of-concept study using an experimental cell-line model, and your method remains to be tested in human blood samples.
2) Methods, section on microvesicles and supernatant preparation (page 4): Regarding microvesicle content, how did you verify this? And for how long was the microvesicle suspensions and supernatants stored before used for experiments?
3) Methods, section on DHM (page 4): I appreciate that you have performed analyses in triplicate, i.e. pooled data from the three different positions. However, I miss information on the analytical variation, i.e. measurement uncertainty on the different morphological parameters and phase contrast values.
Regarding the interpretation of your results and future perspectives:
4) You rightly state in the Introduction that blood cultures have several limitations, including a high false-negative rate and long turn-around times for results. You also stress that distinction between gram-positive and gram-negative bacteria in an early phase may help targeting treatment. It is true that the false negative rate is high for blood cultures, especially if an antibiotic has already been administered, and other reliable biomarkers for bacterial sepsis will certainly be of value. Regarding the long turn-around time for blood cultures, that is also true for final typing of species and strain; however, direct microscopy after gram-staining can be performed quickly and provides information on bacterial morphology, conformation and gram-positivity. I think perhaps the value of your method regarding the determination of gram-positivity is moderate; rather the other parameters that you mention should be stressed.
5) You report different patterns between different strains of bacteria, and it is also true to some extent (figure 1). The results are certainly interesting. However, taken the displayed standard deviations and probable measurement uncertainties into consideration, it could be difficult to get a reliable discrimination between strains in a clinical setting.
6) Continuing on the topic of clinical relevance: Both time since incubation start and bacterial concentration influences the measured cell parameters. This is not unexpected, and it is of interest and provides us with knowledge about pathophysiological mechanisms and the development of T-cell response over time. However, I see this as a considerable challenge to implementation of your method as a diagnostic tool in a clinical setting. Sepsis patients will have 1) different bacterial loads in the blood and 2) will be at varying time points of their infection when admitted to hospital, and information on when the infection started will not always be available (as incubation time may vary).
7) I think you could discuss a little more the somewhat surprising finding that supernatant and microvesicles induced a stronger response than infection with live bacteria. Could it be due to concentrations of live bacteria vs. concentrations of microvesicles or activating factors in the supernatant? Differences in buffers or storage conditions?
Minor comments:
Abstract, page 1, line 19: "Time-lapse DHM was applied […]" The abbreviation DHM should be defined. (I assume it is "digital holographic microscopy"?)
Reviewer 3 Report
Dear Authors,
I was honoured to review your research about sepsis: "Investigating morphological changes of T-lymphocytes after exposure with bacterial determinants for early detection of septic conditions". Investigation T-lymphocyte morphology might provide an important tool for early
identification of bacterial infections. The manuscript has an appropriate research design and the results are presented clearly. I find material and methods well done. I recommend publication.
